# Smartphone Use and Willingness to Pay for HIV Treatment-Assisted Smartphone Applications among HIV-Positive Patients in Urban Clinics of Vietnam

**DOI:** 10.3390/ijerph18041467

**Published:** 2021-02-04

**Authors:** Thu Minh Bui, Men Thi Hoang, Toan Van Ngo, Cuong Duy Do, Son Hong Nghiem, Joshua Byrnes, Dung Tri Phung, Trang Huyen Thi Nguyen, Giang Thu Vu, Hoa Thi Do, Carl A. Latkin, Roger C.M. Ho, Cyrus S.H. Ho

**Affiliations:** 1Bach Mai Medical College, Bach Mai Hospital, Hanoi 100000, Vietnam; minhthu.bmtn@gmail.com; 2Institute for Preventive Medicine and Public Health, Hanoi Medical University, Hanoi 100000, Vietnam; ngovantoan@hmu.edu.vn; 3Institute for Global Health Innovations, Duy Tan University, Da Nang 550000, Vietnam; nguyenthuyentrang46@duytan.edu.vn; 4Faculty of Medicine, Duy Tan University, Da Nang 550000, Vietnam; 5National Hospital of Tropical Diseases, Bach Mai Hospital, Hanoi 100000, Vietnam; doduy.cuong@gmail.com; 6Centre for Applied Health Economics (CAHE), Griffith University, Brisbane, QLD 4222, Australia; s.nghiem@griffith.edu.au (S.H.N.); j.byrnes@griffith.edu.au (J.B.); 7School of Medicine, Griffith University, Gold Coast Campus, Parklands Drive, Southport, QLD 4222, Australia; d.phung@griffith.edu.au; 8Center of Excellence in Evidence-Based Medicine, Nguyen Tat Thanh University, Ho Chi Minh City 700000, Vietnam; giang.coentt@gmail.com; 9Institute of Health Economics and Technology, Hanoi 100000, Vietnam; dothihoa.iheat@gmail.com; 10Bloomberg School of Public Health, Johns Hopkins University, Baltimore, MD 21205, USA; carl.latkin@jhu.edu; 11Department of Psychological Medicine, Yong Loo Lin School of Medicine, National University of Singapore, Singapore 119074, Singapore; pcmrhcm@nus.edu.sg (R.C.M.H.); cyrushosh@gmail.com (C.S.H.H.); 12Institute for Health Innovation and Technology (iHealthtech), National University of Singapore, Singapore 119074, Singapore; 13Department of Psychological Medicine, National University Health System, Singapore 119228, Singapore

**Keywords:** WTP, smartphone, antiretroviral treatment, urban, Vietnam

## Abstract

This study aimed to examine the effectiveness of Human Immunodeficiency Virus (HIV)-assisted smartphone applications in the treatment of HIV/AIDS patients in Vietnam. A cross-sectional study was performed in two urban outpatient clinics in Hanoi from May to December 2019. A simple random sampling method and a structured questionnaire were used to recruit 495 eligible participants and to collect data. Multivariable modified Poisson regression and multivariable linear regression models were employed to investigate the factors associated with the willingness to pay (WTP) and amount of money patients were willing to pay. Approximately 82.8% of respondents were willing to pay for the hypothetical applications, with the mean amount the participants were willing to pay of Vietnam Dong (VND) 72,100/month. Marital status (separate/divorced/widow: Odds ratio (OR) = 1.28, 95% confidence interval (CI) = (1.09; 1.50) and having spouse/partner: OR = 1.18, 95% CI = (1.03; 1.36)) and using health services (OR = 1.03, 95% CI = (1.01; 1.04)) were positively associated with nominating they would be WTP for the app, whereas the duration of antiretroviral treatment (ART) (OR = 0.98, 95% CI = (0.96; 0.99)) had a negative association. The frequency of using health services (β = 0.04, 95% CI = (−0.07; −0.01)) was negatively associated with the amount of WTP. High levels of WTP revealed the feasibility of implementing smartphone-based apps for HIV treatment. This study implied the necessity to consider a co-payment system to reach populations who were in need but where such applications may be unaffordable in lieu of other treatment-associated expenses. Developers also need to pay attention to privacy features to attract single people living with HIV/AIDS and additional measures to initiate people with a long duration on ART into using the applications.

## 1. Introduction

Antiretroviral treatment (ART) has been regarded as the most effective treatment regime to improve patients’ immune system function and to limit the risk of HIV transmission [1,2,3]. The rapid expansion of ART services has reduced the epidemic spread and brought benefits to approximately 21.8 million people living with HIV/AIDS in low- and middle-income nations [4]. However, people living with HIV/AIDS must remain on lifelong treatment and strictly adhere to ART regimens to prevent drug resistance and to achieve treatment success [5,6]. Interventions to support treatment adherence are, therefore, necessary for successful ART programs.

A number of supportive measures have been developed, ranging from directly observed therapy to the design of more complex interventions [7,8]. However, their effectiveness varies across patient groups and settings, which emphasizes the importance of contextual nature in designing supportive measures. Among these, mobile phone-based support has emerged as a promising approach due to its increasingly ubiquitous presence, relatively low cost and the ability to offer highly accessible and adaptable platforms to support treatment adherence [8,9]. For example, Pantoja et al. [10] indicated short message services (SMS) for ART adherence should be adopted as a health strategy in low-income countries. Additionally, Muessig et al. [9] reported the feasibility, acceptability and efficacy of mobile health interventions to support ART adherence among HIV-infected men who have sex with men. A review of 13 studies conducted in eight countries from 2010 to 2016 reported that scheduled-only SMS significantly improved adherence to ART, whilst voice call and triggered SMS did not show any significant improvement [8]. Generally, the majority of respondents expressed satisfaction with mobile phone-based approaches. However, the costs and the willingness to pay (WTP) for mobile phone-based approaches have not been described in any of the above studies.

In Vietnam, the HIV epidemic is still in a concentrated phase in which drug users, commercial sex workers and men who have sex with men are the most-at-risk populations [11]. It is estimated that there are about 320,000 people living with HIV/AIDS and 20% of them are at the advanced HIV stage [12]. With external donor financing and great efforts from the Vietnamese government, free ART services have been scaled up since 2006 [12]. According to the Guideline of HIV/AIDS Treatment and Care issued by the Ministry of Health of Vietnam, all people living with HIV/AIDS are eligible for receiving free ART services, regardless of their CD4 cell count or clinical stage [13]. Until 2016, the coverage of free ART services in Vietnam had reached 48% of people living with HIV/AIDS [14]. However, poor adherence to treatment limits the ability to achieve optimal treatment outcomes and threatens success and the further scaling up of free ART services. Primarily due to the fear of HIV disclosure, stigmatization and certain of socio-economic characteristics; previously reported suboptimal adherence rates vary widely from 29 to 54.5%, [15,16,17,18]. Thus, these associated factors also need to be taken into account when developing supportive interventions.

In 2011, the Vietnam Central Health Information Technology Institute undertook an m-Health project that made mobile technology available to support patients receiving ART services. A year later, Tran et al. [15] conducted a study to measure the feasibility and WTP for that mobile-based application and reported a relatively high level of WTP. However, the functionalities surveyed in that project were basic, including a scheduled reminder, SMS and direct calls by health workers. Presently, smartphones are increasingly available and affordable in Vietnam, with approximately 65% of the population now possessing one. Of which 90% of younger adults aged 18 to 29 own smartphone, the rate of possessing one smartphone of people in group aged 30 to 49 and over 50 are 68% and 29%, respectively [19]. This makes smartphone-based applications a contextually appropriate delivery mechanism and that can go beyond being a basic tool to support HIV self-care management and treatment adherence. Prior simple means to support treatment adherence are, therefore, becoming out of date. As a result, state-of-the-art treatment-assisted applications need to be developed. This study serves as a preliminary investigation of the first longitudinal intervention using smartphones to improve adherence for ART services in Vietnam. We will use baseline data to characterize smartphone use behavior and assess the feasibility, users’ preference and WTP for smartphone-based application for improving ART treatment among people living with HIV/AIDS.

## 2. Materials and Methods

### 2.1. Study Design, Sampling Method, and Data Collection

A cross-sectional study was performed in two urban outpatient clinics in Hanoi, Vietnam: Bach Mai clinic (central level) and Ha Dong clinic (provincial level) from May to December 2019. This study also served as a baseline survey which aimed to examine the effectiveness of HIV-assisted smartphone applications on the treatment of HIV/AIDS patients. According to medical records and face-to-face interviews, patients were recruited who (1) were confirmedly diagnosed with HIV/AIDS; (2) received ART in selected clinics; (3) used smartphones during the study period; (4) agreed to participate and gave their informed consent to participate. A simple random sampling method was used to identify patients for the study. First, we listed all patients who were eligible for the study from the two clinics. After that, we used computer software to randomly select patients on the list and invited them to participate in the study when they visited the clinics for health examination and taking medication. A total of 500 patients were invited, of which the data of 495 patients were eligible and consent was given to participate in the study (99.0% response rate).

Each participant was informed about the study in brief and asked to give their written informed consent. They were invited to go to a private room in the clinic to ensure that their responses were not influenced by other people as well as to protect their privacy. Data from participants were recorded through face-to-face interviews performed by well-trained researchers and not their treating clinicians. Participants’ identifiable information was not collected in the questionnaire in order to facilitate their participation as well as prevent any social desirability bias. Each patient was given an identification code so as to not disclose their private information.

### 2.2. Instruments

A structured questionnaire was developed for the interview. The questionnaire included information about the socio-economic status (sex, age, education, marital status, occupation, and monthly household income), clinical characteristics (HIV stage, initial CD4 cell count, last CD4 cell count, duration of ART, last time forget taking pills and self-reported adherence score and self-reported health status), health behaviors (alcohol use, smoking in the last 30 days, history of drug use and current drug use), and the use of any health service in the last 12 months. The household income data were used to divide the sample into five quintiles: first (VND ≤ 3 million/month), second (VND > 3 million and VND ≤ 4 million/month), third (VND > 4 million and ≤5 million/month), fourth (VND >5 million and VND ≤ 7 million/month) and fifth (VND > 7 million/month). The self-reported adherence score was assessed by using the visual analogue scales with a score ranging from 0 “complete non-adherence” to 100 “complete adherences”.

For smartphone use, we asked patients to report their duration of using a smartphone; time spent using a use per day; use of health-related smartphone applications (apps) in the last 30 days; time spent on health-related apps per day; the functionality of health-related apps used; time when using apps; perceived usefulness of health-related apps (with a range score from 0 “Completely useless” to 100 “Completely useful”) and satisfaction with health-related apps (with a range score from 0 “Complete dissatisfaction” to 100 “Complete satisfaction”).

Moreover, we described an application that can be used to support them in HIV treatment and asked them to express whether they would be willing to pay for this app (yes or no). For those who are willing to pay for the app, the maximum amount of willingness to pay per month for using this app was also evaluated by asking them to respond the question “What is the maximum amount of money that you will be willing to pay for this application?” (in thousands of VND). The questionnaire for using a health-related application is provided in Table A1 (Appendix A).

### 2.3. Data Analysis

Statistical significance was detected if a *p*-value was less than 0.05. Stata software version 15.0 was used for the data analysis. Two-tailed chi-squared and Mann–Whitney tests were performed to examine the difference in demand, WTP and amount of WTP for HIV-assisted smartphone applications according to different socio-economic, clinical and behavioral characteristics. A multivariate modified Poisson regression model was then performed to investigate the factors associated with willingness to pay for smartphone apps (as the dependent variable) (model 1). Meanwhile, we transformed the data for the amount of money that participants were willing to pay to the logarithm form due to the non-normal distribution of the data. Then, we conducted a multivariable linear regression model to determine the associated factors with the amount of WTP (as a dependent variable for model 2). Independent variables for both models included priori-defined variables such as socio-economic status (sex, age, marital status, education level, employment status, monthly household income quintiles), clinical characteristics (HIV stage, initial CD4 cell count, last CD4 cell count, duration of ART, last time forget taking pills and self-reported adherence score and self-reported health status), health behaviors (alcohol use, smoking in the last 30 days, history of drug use and current drug use), and use of any health service in the last 12 months. The variance inflation factor (VIF) analysis was performed to examine collinearity among the variables. No collinearity was found. Stepwise forward selection strategies were employed together with the regression models to produce the reduced models. A *p*-value of 0.2 of the log-likelihood test had been used in previous studies among HIV/AIDS patients as a threshold for including the variables for the final models [20,21]. The results of age and sex subgroup analyses were presented in the Table A2 and Table A3 (Appendix A).

### 2.4. Ethical Consideration

The protocol of this study was approved by the institutional review board of Hanoi Medical University (Code: 18NCS17/HDDDDHYHN).

## 3. Results

The baseline characteristics of the participants are described in Table 1. Among 495 patients, the majority were male (57.0%) and had a spouse/partner (69.1%). The mean age was 37.8 (Standard deviation (SD) = 6.6) years old. Most patients had high school education or above. The average monthly household income was VND 5361.0 thousand (SD = 7358.1). The proportions of alcohol users, current smokers and current drug users were 50.5, 32.5 and 0.8%, respectively. Most of the patients were at the asymptomatic stage (95.5%), and 86.1% reported that they did not forget to take any pill during ART treatment.

Table 2 reveals that the mean duration of smartphone use among our sample was 28.7 (SD = 21.2) months. Only 7.3% had ever used health-related smartphone applications (apps) in the last 30 days, with medication reminders as the most common function (75.0%). Most app users utilized the app in the early morning (58.3%) and evening (16.7%). The majority of HIV patients (82.8%) were willing to pay for the apps with the mean amount of WTP of VND 72.1 thousand/month (SD = 36.3).

The WTP and amount of WTP regarding different socio-demographic and clinical characteristics are presented in Table 3. Significant differences in WTP for smartphone apps were only found among marital groups (*p* < 0.05). Meanwhile, no difference was observed in the other characteristics.

Figure 1 illustrates that among those who were willing to pay for the smartphone app, the median WTP amount for this app was VND 60 thousand, with less than 2.5% of patients willing to pay VND 100 thousand or more.

Factors associated with the WTP and amount of WTP for HIV-assisted smartphone apps are shown in Table 4. Separated/divorce/widow or having spouse/partner and using health service in the last 12 months were associated with WTP for the apps. Meanwhile, the higher duration of ART was negatively associated with WTP for the apps (OR = 0.98, 95% CI = 0.96–0.99).

Regarding the amount of WTP, a higher age or using health services in the last 12 months were negatively related to the amount of WTP. Particularly, a person by aged one more year than the mean will pay 1% lower for the app; and compared to those who did not use health services in the past 12 months, the users of health services were willing to pay 4% lower for the app (Table 4).

## 4. Discussion

This study examined the use of smartphone-based health-related apps and WTP for them among HIV patients in urban clinics of Vietnam. We found that only a tiny proportion of respondents were using health-related apps whilst the vast majority of them were willing to pay for an HIV treatment-assisted application. The study also revealed that the marital status and using health services in the last 12 months were positively associated with a WTP, whereas the duration of ART had a negative association. Regarding the amount for WTP, the frequency of using health services in the last 12 months was negatively associated with the WTP for the app. Our findings provide useful insights into the WTP and suggest the potential for developing HIV treatment-assisted apps in Vietnam.

Our findings confirmed the potential for the use of mHealth in assisting patients’ treatment adherence in resource-limited settings [22,23,24]. In the context of Vietnam, a small percentage of respondents who were using health-related apps was comparable to that of a survey on smartphone-based vaccination application (5%) [25]. However, the proportion of participants willing to pay for an app in this study was higher than the previous studies related to vaccination (79.1%) [25], ART adherence support (63.5%) [15], and smoking cessation (26.8%) [26]. Regarding the amount of money, the average amount for WTP found in this study was higher than that reported in a prior ART adherence-support application (VND 51,000 or United States dollar (USD) 2.5 per month) [15]. There are several possible explanations for this discrepancy. First, the previous study was conducted seven years ago; therefore, a higher amount of WTP in our study might account for the impact of inflation in later years. Second, the existing functionalities of apps used in previous studies were basic, including text messaging, voice messaging and direct calls from health workers [15,27], whilst when we asked participants about the amount of money, they were WTP for an application offering a broad complement of functionalities, specifically to assist with HIV treatment. Therefore, a higher price might be paid for additional functions. The high level of WTP for the HIV treatment-assisted application, along with the poor features of existing ones suggest that well-developed, highly interactive apps would be welcomed and utilized in the HIV-positive population.

A strong association was observed between “using health services in the last 12 months” with both WTP and the amount WTP for the application. On the one hand, this independent variable was positively associated with a WTP. We presumed that those who used health services experienced the current cumbersome procedures for diagnosis and treatment in Vietnam. Hence, they valued integrating healthcare service information systems in which their medical records and prescriptions could be managed more conveniently; the benefits of updating health-relative news and information as well as contact with nearby clinics [28,29,30]. On the other hand, the frequency of using health services in the last 12 months was negatively associated with the WTP for the apps. A probable explanation is that those patients had already spent a large number of their health budgets while using services in the last 12 months, which may have led to the financial burden. Consequently, they may be more considerate in spending on additional HIV-related services [31]. This finding suggests that a co-payment method could help in reaching sub-populations who were in need but otherwise, might not afford it in light of other healthcare expenses. Given the financial concerns of those who have already spent money on other healthcare services in the last 12 months, findings also suggest that vendors of the application could consider options within the app to encourage those who have previously incurred health service costs as well as building loyalty to the app itself; for example, promoting a discount program for frequently using the app, a reward for referring friends, etc.

Upon analysis, those who are or had been married (i.e., married, separate, divorced or widow) or have a partner were more willing to pay for the application compared to their single counterparts. Since family members are often involved in ART treatment support by providing daily medication reminders for the people living with HIV [32,33,34], these groups of patients may be aware of the benefit from reminders and other treatment-assisted functions. By contrast, this result may also be due to increased concern about privacy loss and the stigma of HIV disclosure among single patients. Indeed, single people living with HIV/AIDS have previously been found to be less likely to disclose their HIV situation because of experienced stigmatization driven by discrimination, fear of being blamed or judged [15,18]. These above concerns might act as deterrents to single patients’ WTP for the application. This implies that the design of prospective apps targeting an HIV-positive population should tackle the variety of demands of different groups. Accordingly, we recommend developers to pay attention to improving not only various functionalities of the app but also to privacy features and ensuring that the app is discreet with regards to the name, logo, layout and design of the app.

This study also revealed that the duration of ART was negatively associated with the WTP for the apps. This might be explained by the fact that over longer periods of ART, people living with HIV/AIDS require fewer reminders and external support as they have already formed a habit of taking pills regularly [35]. However, the associated problem is that their overconfidence might also lead to carelessness and losing adherence over time. Indeed, evidence in Vietnam indicates that patients with a longer duration on ART are at increased probability of forgetting to take pills [36]. As one of the primary functionalities was to facilitate treatment adherence, it would be promising if these group of people living with HIV/AIDS were initiated into using the apps, potentially through preferential pricing options, government co-payment either in perpetuity or for an initial time period. Once they experienced and perceived the usefulness of smartphone-based applications, the likelihood of WTP for it might be increased.

There are several limitations to this study that should be noted. First, the analysis is based on a self-report survey that may involve recall bias. However, to minimize this type of bias related to a critical input “using health-related applications”, we asked participants to report their utilization “in the last 30 days”. Second, our study had a common weakness of the cross-sectional study design in which the causal-effect relationship between dependent variables and independent variables was not identified. Analyzing the follow-up data in the future longitudinal study is expected to elucidate this issue. Third, the newly developed questionnaire for using a health-related application that we used in this study had not been validated elsewhere. Finally, the sample was drawn from two outpatient clinics in a metropolis; it, therefore, was not representative of people living with HIV/AIDS in Vietnam as a whole. Nevertheless, using a simple random sampling method enables our findings to apply to urban areas’ target population.

## 5. Conclusions

Smartphone-based applications present as promising initiatives to enhance treatment adherence, patients’ engagement and to improve provider–patient relationships. This study provides evidence to support the potential use of smartphone-based applications among people living with HIV/AIDS in Vietnam. Our findings showed that the majority of participants were willing to pay for HIV treatment-assisted apps and identified marital status, using health services in the last 12 months and the duration of ART were factors associated with WTP as well as the amount of WTP for HIV treatment-assisted apps in urban clinics in Vietnam. In addition to the feasibility of implementing smartphone-based applications for HIV treatment, this study also suggests the consideration of a co-payment system to reach populations who are in need but may find the app unaffordable due to associated healthcare expenses. At the same time, developers also need to pay attention to privacy features to attract single people living with HIV/AIDS and initiate people with long-duration on ART into using the apps. Future research should consider exploring the different revenue models that could be adopted by this app.

## Figures and Tables

**Figure 1 ijerph-18-01467-f001:**
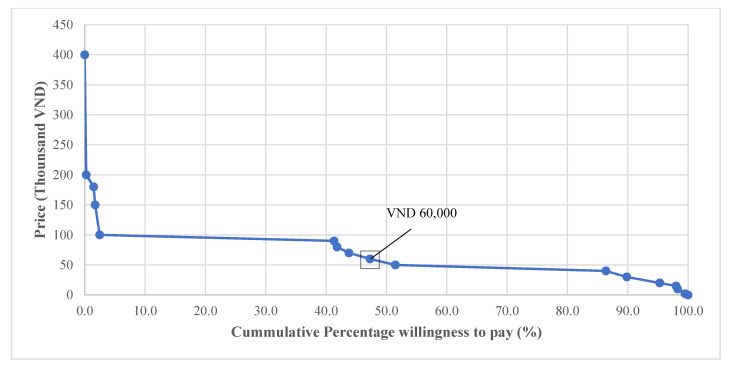
Cumulative percentage of the amount of WTP.

**Table 1 ijerph-18-01467-t001:** Socio-demographic and clinical characteristics.

Characteristics	*n*	%
**Sex (*n* = 495)**		
Male	282	57.0
Female	213	43.0
**Marital status (*n* = 495)**		
Single	84	17.0
Separate/divorce/widow	69	13.9
Having spouse/partner	342	69.1
**Education (*n* = 495)**		
Elementary school or under	48	9.7
Secondary school	152	30.7
High school	169	34.1
Above high school	126	25.5
**Employment status (*n* = 495)**		
Unemployed	27	5.5
White-collar workers	45	9.1
Blue-collar workers/farmers	94	19.0
Self-employed	34	6.9
Others	295	59.6
**Self-reported HIV stage (*n* = 487)**		
Asymptomatic	465	95.5
Symptomatic	22	4.5
**Last time forget taking an HIV pill (*n* = 483)**		
Never	416	86.1
Last week	13	2.7
Last 1–2 week	6	1.2
Last 2–4 week	3	0.6
Last 1–3 months	14	2.9
Last more than 3 months	16	3.3
Does not remember	15	3.1
**Uses alcohol (*n* = 495)**	250	50.5
**Smoked in the last 30 days (*n* = 495)**	161	32.5
**History of drug use (*n* = 495)**	115	23.2
**Current drug use (*n* = 495)**	4	0.8
**Using health service in the last 12 months (*n* = 495)**	98	19.8
	**Mean**	**SD**
Age (*n* = 495)	37.8	6.6
Monthly household income (thousand VND) (*n* = 495)	5361.0	7358.1
CD4 first (*n* = 481)	271.5	239.1
CD4 last (*n* = 478)	529.3	267.8
Duration in *ART* (years) (*n* = 488)	6.2	2.8
Adherence score (0–100) (*n* = 476)	94.0	10.3
Self-reported health status (0–100) (*n* = 495)	77.9	16.2

**Table 2 ijerph-18-01467-t002:** Health-related smartphone application use, demand and willingness to pay (WTP).

Characteristics	*n*	%
**Using health-related application(s) in the last 30 days (*n* = 495)**	36	7.3
**Function used the most (*n* = 36)**		
Smoking cessation	1	2.8
Physical activity	8	22.2
Remind medication	27	75.0
**Day session when using apps (*n* = 36)**		
Early morning	21	58.3
Morning	3	8.3
Noon	1	2.8
Afternoon	2	5.6
Evening	6	16.7
Before night sleep	3	8.3
**WTP for smartphone apps (*n* = 495)**	410	82.8
	**Mean**	**SD**
Duration of smartphone use (month) (*n* = 495)	28.7	21.2
Time of smartphone use per day (hours) (*n* = 36)	3.4	1.8
Time of health-related application use per day (minutes) (*n* = 36)	14.3	11.0
Perceived usefulness of health-related apps (0–100) (*n* = 36)	85.3	13.2
Satisfaction with health-related apps (0–100) (*n* = 36)	86.9	12.8
Amount of WTP (thousand VND/month) (*n* = 404)	72.1	36.3

**Table 3 ijerph-18-01467-t003:** WTP and amount of WTP according to the different socio-demographic and clinical categories.

Characteristics	WTP for Smartphone App			Amount of WTP		*p*-Value
	*W*	%	*p*-Value	Mean	SD	
**Sex**						
Male	221	79.8	0.06	74.6	39.3	0.18
Female	182	86.3		68.9	32.0	
**Marital status**						
Single	62	74.7	0.049	82.6	51.0	0.10
Separate/divorce/widow	61	89.7		70.2	26.2	
Having spouse/partner	280	83.1		70.1	33.8	
**Education**						
Elementary school or less	38	79.2	0.36	74.2	40.1	0.48
Secondary school	119	79.3		67.3	28.8	
High school	139	83.2		73.3	31.1	
Above high school	107	87.0		75.1	47.0	
**Job**						
Unemployed	22	81.5	0.79	68.6	30.8	0.90
White-collar workers	35	81.4		69.4	33.3	
Blue-collar Workers	77	81.9		72.5	35.0	
Self-employed	30	90.9		73.2	37.8	
Others	239	82.1		72.5	37.5	
**HIV stage**						
Asymptomatic	379	82.8	0.91	72.3	36.5	0.44
Symptomatic	18	81.8		67.2	33.6	
**Last time forgot taking pill**						
Never	339	82.5	0.64	73.0	37.5	0.83
Last week	11	84.6		76.4	28.0	
Last 1–2 week	6	100.0		61.7	19.4	
Last 2–4 week	3	100.0		66.7	28.9	
Last 1–3 months	11	78.6		72.7	26.1	
more than 3 months	11	68.8		63.2	37.8	
Does not remember	11	78.6		-	-	
**Use Alcohol**	95	79.2	0.26	72.7	27.0	0.35
**Smoke last 30 days**	128	80.5	0.40	71.0	32.1	0.96
**History of drug use**	88	77.9	0.13	72.3	34.7	0.87
**Household monthly income (VND/month)**						
≤3 million	129	83.2	0.83	72.4	44.7	0.55
3 to 4 million	50	78.1		69.2	26.3	
4 to 5 million	91	84.3		70.5	30.3	
5 to 7 million	65	84.4		79.6	39.1	
>7 million	68	81.0		68.3	27.8	

**Table 4 ijerph-18-01467-t004:** Associated factors with WTP and the amount of WTP.

Characteristics	WTP for Smartphone App		Natural Log of the Amount of WTP	
	OR	95% CI	Β	95% CI
**Socio-demographic Characteristics**				
Age (years)			−0.01 **	−0.02; 0.00
Marital status (single-ref)				
Separate/divorced/widow	1.28 ***	1.09; 1.50		
Having spouse/partner	1.18 ***	1.03; 1.36		
Education (Elementary school or less-ref)				
Secondary school	1.00	0.85; 1.19		
High school	1.07	0.90; 1.26		
Above high school	1.16 *	0.99; 1.37		
**Clinical Characteristics**				
Using health services in the last 12 months (Yes vs. No -ref)	1.03 ***	1.01; 1.04	−0.04 **	−0.07; −0.01
Duration of ART (years)	0.98 **	0.96; 0.99		
Using alcohol (Yes vs. No-ref)	0.94	0.87; 1.02		

*** *p* < 0.01, ** *p* < 0.05, * *p* < 0.1.

## Data Availability

The data presented in this study are available on request from the corresponding author. The data are not publicly available due to privacy.

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
