# Peer review of "Smartphone Use and Willingness to Pay for HIV Treatment-Assisted Smartphone Applications among HIV-Positive Patients in Urban Clinics of Vietnam"

_ijerph, 2021, doi:10.3390/ijerph18041467_

Round 1
Reviewer 1 Report
Regarding the app, it is useful to provide some additional information on the potential/different services it could offer, also to estimate the price that clients would be willing to pay (value for money).
It is also useful to specify which question has been asked about the revenue model adopted by the app (whether for example a single payment to download, subscription, etc.)
It is necessary to specify better which variables have been included in the Regression, as dependent variable (e.g. intention to pay or price of the app) and as independents ones.
It is necessary to present a table on the main results of the Regression and in particular to describe the representativeness of the β different variables. An analysis of collinearity is useful.
In the conclusion, it is useful to report some indication of the different revenue models that could be adopted by this app.
Author Response
Regarding the app, it is useful to provide some additional information on the potential/different services it could offer, also to estimate the price that clients would be willing to pay (value for money).
It is also useful to specify which question has been asked about the revenue model adopted by the app (whether for example a single payment to download, subscription, etc.)
Response:
Thank you for your valuable feedback. We added the questionnaire that used for smartphone application section in the appendix.
Position: Line 329
It is necessary to specify better which variables have been included in the Regression, as dependent variable (e.g. intention to pay or price of the app) and as independents ones.
Response:
Thank you for your comment.
We included independent variables in the Data analysis part.
“ Independent variables for both models included priori-defined variables such as so-cio-economic status (gender, age, marital status, education level, employment status, monthly household income quintiles), clinical characteristics (HIV stage, initial CD4 cell count, last CD4 cell count, duration of ART, last time forget taking pills and self-reported adherence score and self-reported health status), health behaviors (alcohol use, smoking in the last 30 days, history of drug use and current drug users), and use of any health service in the last 12 months.
Position: Lines 169-175
It is necessary to present a table on the main results of the Regression and in particular to describe the representativeness of the β different variables.
Response:
Regarding the representativeness of the β different variables, they are equivalent to Coef. in the regression result table. According to your comment, we changed Coef. into β.
Position: Table 4 – line 220
An analysis of collinearity is useful.
Response:
We ran Variance inflation factor analysis to examine collinearity among variables. Yet, none of collinearity was found. We also added these sentences in the data analysis part.
Position: Lines 175-176
In the conclusion, it is useful to report some indication of the different revenue models that could be adopted by this app.
Response:
Regarding the different revenue models, we thought it was beyond the realms of this study. However, thanks to your comment, we add it as a suggestion for future research.
Position: Lines 328 to 329
Reviewer 2 Report
This paper is a topic of interest to the researchers in related areas. It aims to examine the effectiveness of HIV-assisted smartphone applications on 36 the treatment of HIV/AIDS patients in Vietnam. Overall, I think this paper is well organized and its presentation is good. Before it can be accepted for publication, I suggest the authors make the following revisions. In conclusions, the authors can describe future possibly extended research. A native speaker and a more careful proofreading should be consideredAuthor Response
RESPONSE TO REVIEWER 2 COMMENTS:
Reviewer 2:
This paper is a topic of interest to the researchers in related areas. It aims to examine the effectiveness of HIV-assisted smartphone applications on the treatment of HIV/AIDS patients in Vietnam. Overall, I think this paper is well organized and its presentation is good. Before it can be accepted for publication, I suggest the authors make the following revisions. In conclusions, the authors can describe future possibly extended research. A native speaker and a more careful proofreading should be considered.
Response:
Thanks to your feedback, we added indication of the different revenue models that could be adopted by this app as a suggestion for future research.
Position: Line 328 to 329
Reviewer 3 Report
IJERPH-1085566 presents results from a study related to smartphone usage. While some parts of this manuscript were interesting, other areas could be improved. I hope the authors consider my feedback.
MAJOR COMMENTS
- Lines 104-108 and 113-115: This reviewer is confused about the mentioning of this research being part of a randomized controlled intervention, when the study is listed as being cross-sectional. Please delete text that is not relevant to the dissemination of this particular research to reduce reader confusion or clarify herein.
- Introduction: It was listed that about 65% of persons in Vietnam have smartphones. While this metric helps to support the significance of the study, it may be of interest to further look at different demographics (e.g., age, sex, SES, etc.) and the proportions of these persons that have smartphones. Does this also corroborate with your population of interest? This helps to better establish specificity in the study sample.
- Line 139-140: Household income as poorest, poor, middle, etc. need better operationalization. List the specific income levels that correspond with this character definition.
- Statistical Analysis: Please list (specify) all of the variables that were included in the stepwise procedure so that we know the start (statistical analysis) and end (Results) points of the models.
- Statistical Analysis: You may also want to examine sub-group analyses (e.g., sex, age) as an appendix.
MINOR COMMENTS
- You may want to reduce the number of abbreviations used in the paper for improved readability. I believe there are no word restrictions in IJERPH.
- Line 74: Are SMS and text messages generally the same? If the authors believe so, then consider using SMS throughout the manuscript.
- Lines 115-118: How were recruitment criteria confirmed? Self-report?
- Line 124: Confirm “written” informed consent in the text.
- Lines 140-142: Can more information be included here?
- Section 2.2: Is there any validity information that can support the use of the questionnaires for research purposes?
- Statistical Analysis: Please list the statistical software used. Please also clarify “modified” Poisson regression in the text.
- The spacing between the tables and text could be improved for readability.
- Data elements: Please make sure that all tables and figure stand-alone (e.g., abbreviations are defined in table notes).
- Discussion: Be careful about the financial aspect of the smart phone app such that it should avoid a business perspective.
- Please include specific results (i.e., statistics) in the abstract.
- Make any changes to the abstract that align with those from the text.
Author Response
RESPONSE TO REVIEWER 3 COMMENTS
IJERPH-1085566 presents result from a study related to smartphone usage. While some parts of this manuscript were interesting, other areas could be improved. I hope the authors consider my feedback.
MAJOR COMMENTS
Lines 104-108 and 113-115: This reviewer is confused about the mentioning of this research being part of a randomized controlled intervention, when the study is listed as being cross-sectional. Please delete text that is not relevant to the dissemination of this particular research to reduce reader confusion or clarify herein.
Response:
Thank you for your valuable feedback.
We deleted ambiguous words as your comment.
Position: Line 113
Introduction: It was listed that about 65% of persons in Vietnam have smartphones. While this metric helps to support the significance of the study, it may be of interest to further look at different demographics (e.g., age, sex, SES, etc.) and the proportions of these persons that have smartphones. Does this also corroborate with your population of interest? This helps to better establish specificity in the study sample.
Response:
More information about the rate of possessing one smartphone of people in 3 groups: from 18 to 29, from 30 to 49 and over 50 was reported in the introduction.
Position: lines 101-103.
Line 139-140: Household income as poorest, poor, middle, etc. need better operationalization. List the specific income levels that correspond with this character definition.
Response:
We revised the manuscript and divided household income into 5 quintiles: first (≤ 3 million VND/month), second (>3 million VND and ≤ 4 million VND/month), third (> 4 million VND and ≤5 million VND/month), fourth (> 5 million VND and ≤7 million VND/month) and fifth (> 7 million VND).
Position: Lines 140-142
Statistical Analysis: Please list (specify) all of the variables that were included in the stepwise procedure so that we know the start (statistical analysis) and end (Results) points of the models.
Response:
Thanks to your comment, we provided more details in the data analysis as following:
“A multivariate modified-Poisson Regression model was then performed to investigate the factors associated with willingness to pay for smartphone apps (as dependent var-iable) (model 1). Meanwhile, we transformed data for the amount of money that par-ticipants were willing to pay to the logarithm form due to non-normal distribution data. Then, we conducted a Multivariable Linear Regression model to determine the associ-ated factors with amount of WTP (as a dependent variable for model 2). Independent variables for both models included priori-defined variables such as socio-economic status (gender, age, marital status, education level, employment status, monthly household income quintiles), clinical characteristics (HIV stage, initial CD4 cell count, last CD4 cell count, duration of ART, last time forget taking pills and self-reported ad-herence score and self-reported health status), health behaviors (alcohol use, smoking in the last 30 days, history of drug use and current drug users), and use of any health ser-vice in the last 12 months”.
Position: Lines 163-175
Statistical Analysis: You may also want to examine sub-group analyses (e.g., sex, age) as an appendix.
Response:
We did analyze sub-groups and presented in the table 4.
Position: Line 220
MINOR COMMENTS
You may want to reduce the number of abbreviations used in the paper for improved readability. I believe there are no word restrictions in IJERPH.
Line 74: Are SMS and text messages generally the same? If the authors believe so, then consider using SMS throughout the manuscript.
Response:
Thanks to you comment, we used SMS instead of text messages. We also reduced the number of abbreviations such as: MSM (men who have sex with men) or PLWH (people living with HIV/AIDS)
Lines 115-118: How were recruitment criteria confirmed? Self-report?
Response:
According to medical records, we recruited those who met the first two criteria: 1) were confirmedly diagnosed with HIV/AIDS; 2) received ART in selected clinics. Then, well-trained researchers conducted face-to-face interviews and asked whether the patients were using smartphone and willing to participate in this study.
Position: Line 116
Line 124: Confirm “written” informed consent in the text.
Response:
We added “written” informed consent as your recommendation.
Lines 140-142: Can more information be included here?
Response:
We provided details of questionnaire in the appendix.
Position: Line 333
Section 2.2: Is there any validity information that can support the use of the questionnaires for research purposes?
Response:
Since questionnaire for using health-related application had not been validated elsewhere, we stated this limitation in the discussion section.
“Third, the newly developed questionnaire for using health-related application that we used in this study had not been validated elsewhere”.
Position: Lines 306-308
Statistical Analysis: Please list the statistical software used. Please also clarify “modified” Poisson regression in the text.
Response:
Stata software version 15.0 was used for data analysis.
Lines 160-161
We also provided more information regarding regression models from lines 164 to 169.
The spacing between the tables and text could be improved for readability.
Response:
Thank you for your comment. We formatted the manuscript to make it easier to follow.
Data elements: Please make sure that all tables and figure stand-alone (e.g., abbreviations are defined in table notes).
Response:
Thank you for your comment. We formatted the manuscript to make it easier to follow.
Discussion: Be careful about the financial aspect of the smart phone app such that it should avoid a business perspective.
Response:
Thank you for your comment. We deleted the clause “although launching smartphone-based apps might provide financial benefits for application vendors”, the statement in revised version was “This finding suggests that a co-payment method could help in reaching sub-populations who were in need but otherwise, might not afford it in light of other healthcare expenses”.
Position: Lines 266 to 268
Please include specific results (i.e., statistics) in the abstract.
Make any changes to the abstract that align with those from the text.
Response:
Thanks to your feedback, we added OR and 95% CI in the abstract.
Positions: Lines 41-45

Reviewer 4 Report
I have reviewed the article sent for review. I find it interesting as the use of computer applications on smart phones can significantly improve many aspects of medical practice. One of them is treatment adherence.
I don't know if the application developed by the authors can also respond to warning signs of illness that can filter out the doctors attending the patient. I think the authors should reflect on this aspect. Likewise, it would be interesting to know what the results are in the population studied, which has a cultural and social connotation, with respect to other experiences on the same subject carried out in other countries.
Author Response
RESPONSE TO REVIEWER 4 COMMENTS
I have reviewed the article sent for review. I find it interesting as the use of computer applications on smart phones can significantly improve many aspects of medical practice. One of them is treatment adherence.
I don't know if the application developed by the authors can also respond to warning signs of illness that can filter out the doctors attending the patient. I think the authors should reflect on this aspect.
Response:
Thank you for your comments that are very useful to improve our manuscript.
Functions of smartphone app that used in this study included: assisting patients in taking pill reminder, managing their medical records, updating health-related information, and contacting nearby physicians and health facilities. In the appendix, question number S13, we asked the participants if they were willing to pay for an HIV/AIDS treatment-assisted app with mentioned functions.
Position: Table line 329
Likewise, it would be interesting to know what the results are in the population studied, which has a cultural and social connotation, with respect to other experiences on the same subject carried out in other countries.
Response:
Thank you for your comment. However, because there has been limited number of papers studying WTP for HIV/AIDS treatment-assisted application, we cannot find and compare results of the same subject in other countries.
Round 2
Reviewer 3 Report
The authors have done a nice job addressing my previous critiques. I hope the authors consider some additional feedback for their manuscript.
- Line 160: Should be “(See more in the Appendix).” Be sure to clean up any minor typos throughout (e.g., line 226; *p<0.).
- Line 181-182: Insert a citation to support a p-value of 0.2 here.
- Although the authors stated in their reply that they conducted stratified analyses for sex and age, this reviewer does not see such analyses in Table 4 or the statistical analysis section.
Author Response
Response to the Reviewer 3
Thank you for your constructive feedback.
Line 160: Should be “(See more in the Appendix).” Be sure to clean up any minor typos throughout (e.g., line 226; *p<0.).
Response:
We have carefully revised the manuscript and correct all errors.
Line 181-182: Insert a citation to support a p-value of 0.2 here.
Response:
The threshold (p-value ≤0.2) had been used in previous studies among HIV/AIDS patients. We cited 2 papers that used the same threshold.
Position: Lines 182-183
Although the authors stated in their reply that they conducted stratified analyses for sex and age, this reviewer does not see such analyses in Table 4 or the statistical analysis section.
Response:
We added 2 tables “Associated factors with WTP and amount of WTP stratified by gender and age groups” in the appendix A2 and A3.
Position: Lines 342 - 345
